# Optical Temperature Sensor Based on Polysilicon Waveguides

**DOI:** 10.3390/s22239357

**Published:** 2022-12-01

**Authors:** Xinru Xu, Yuexin Yin, Chunlei Sun, Lan Li, Hongtao Lin, Bo Tang, Peng Zhang, Changming Chen, Daming Zhang

**Affiliations:** 1State Key Laboratory of Integrated Optoelectronics, College of Electronic Science and Engineering, Jilin University, Changchun 130012, China; 2Key Laboratory of 3D Micro/Nano Fabrication and Characterization of Zhejiang Province, School of Engineering, Westlake University, Hangzhou 310024, China; 3Institute of Advanced Technology, Westlake Institute for Advanced Study, Hangzhou 310024, China; 4State Key Laboratory of Modern Optical Instrumentation, College of Information Science and Electronic Engineering, Zhejiang University, Hangzhou 310027, China; 5Institute of Microelectronics of the Chinese Academy of Sciences, Beijing 100029, China

**Keywords:** temperature sensors, polysilicon, dual-microring resonator, AMZI

## Abstract

Traditional temperature detection has limitations in terms of sensing accuracy and response time, while chip-level photoelectric sensors based on the thermo-optic effect can improve measurement sensitivity and reduce costs. This paper presents on-chip temperature sensors based on polysilicon (p-Si) waveguides. Dual-microring resonator (MRR) and asymmetric Mach–Zehnder interferometer (AMZI) sensors are demonstrated. The experimental results show that the sensitivities of the sensors based on AMZI and MRR are 86.6 pm/K and 85.7 pm/K, respectively. The temperature sensors proposed in this paper are compatible with the complementary metal-oxide-semiconductor (CMOS) fabrication technique. Benefitting from high sensitivity and a compact footprint, these sensors show great potential in the field of photonic-electronic applications.

## 1. Introduction

Silicon photonics are a promising solution for large-scale and high-performance photonic integrated circuits (PICs) with a growing amount of applications in the optical interconnections [1,2], programmable photonics processors [3,4], and optical sensors [5,6,7]. Integrated silicon photonics sensors especially, have attracted great attention in the fields of environmental monitoring [8], industrial production [6,9], medical diagnosis [10], and chemical analysis [11], due to their high sensitivity, compact footprint, and mass production capabilities [12,13]. Besides high-performance optical sensors, readout equipment and interrogators are also crucial for lab-on-chip (LoC) photonic sensors [5,7].

Nowadays, most silicon photonic devices and circuits are demonstrated on the silicon-on-insulator (SOI) platform [14,15,16]. Crystalline silicon (c-Si) is the most widely used material due to its low optical losses and excellent electronic properties [17]. However, it is challenging to achieve three-dimensional (3D) multilayer PICs and electronic-photonics integrated circuits (EPICs) based on c-Si due to the extreme difficulty in full wafer bonding and polishing. In that case, low-temperature deposited materials such as hydrogenated amorphous silicon (a-Si:H) [18,19], silicon nitride (SiN) [20,21,22,23,24,25], or polysilicon (p-Si) [26] have been deposited and pattern above SOI wafers for multilayer PICs and EPICs. However, the low charge mobility in a-Si:H and SiN thin films limits their use as active components such as electro-optic modulators. P-Si, a collection of single-crystal silicon grains separated by grain boundaries, shows relatively low-loss and similar mobility to c-Si [27]. Additionally, the p-Si thin film transistor (TFT) has the merits such as high field effect mobility, high integration and high speed, high-definition display, low power consumption, and self-aligned structures [28,29,30]. With these good characteristics, p-Si is a promising material for realizing multilayer active photonic devices and EPICs [31,32]. For a reliable on-chip system, a temperature monitor is necessary. Optical sensors have attracted lots of attention because of their high sensitivity, compact footprint, and property of anti-electromagnetic interference. To improve the reliability and performance of 3D PICs, in this paper, we demonstrate two kinds of p-Si temperature sensors based on the dual-microring resonator (MRR) and asymmetric Mach–Zehnder interferometer (AMZI). The temperature sensitivities of the sensors are 87 pm/K and 85 pm/K, respectively. Both of these sensors are suitable for temperature measurement of 3D PICs and EPICs.

## 2. Design, Fabrication, and Characterization

The devices were fabricated with the 180 nm p-Si photonic multi-project-wafer (MPW) process at the Institute of Microelectronics of the Chinese Academy of Sciences (IMECAS). The 220 nm thick p-Si was deposited on the SiO_2_ buffer at 2 μm of thickness at 620 °C. The sensor is designed for C-band operation. For single-mode propagation, the geometry of the waveguide is 500 nm × 220 nm. Deep ultraviolet (DUV) photolithography was employed to define the waveguide patterns, followed by inductively coupled plasma (ICP) etching of silicon. To simplify the fabrication process, we chose one-step etching of 150 nm for both grating couplers and ridge waveguides. A 1 μm thick SiO_2_ upper cladding was deposited on the waveguides. 

### 2.1. The Structure of MRR

MRRs are usually compact with a typical radius of several micrometers. The narrow linewidth of resonance dip is attractive for highly sensitive optical sensors [33]. However, the large detection range means a small ring radius, which results in increased bending loss in the ring. To overcome the trade-off between detection range and sensitivity, we adopt a dual-MRR structure with different circumferences and a different free spectral range (FSR), as shown in Figure 1. As a result, the transmittance reaches the minimum only at the overlapped resonant peaks for each of the ring resonator.

Our dual-MRR sensor consists of a ring (MRR1) and a racetrack (MRR2) resonator. The gap between the ring and bus waveguide is 300 nm. The racetrack resonator consists of a directional coupling (DC) coupling region. For the coupling region, the gap and the length are 400 nm and 10 μm, respectively. The radii of the ring and racetrack are 30 μm and 40 μm, respectively. A broadband tunable laser system (Santec Full-band TSL-550) covering the ultra-wide tuning range of 1260 nm to 1630 nm was used to characterize the fabricated devices. The resolution is 1 pm for MRR, narrow linewidth components, in the measurements. This system combines up to three tunable lasers (TSL-550) with an optical switch module (OSA-110). The light from output grating is measured through an optical power meter (MPM-210H). The light from laser and devices is coupled into the devices through a vertical fiber coupling system with a polarization controller. The minimum loss of the reference grating is ~10.42 dB for the fundamental transverse electric (TE) mode. The spectrum of the dual-MRR sensor from 1543 nm to 1568 nm at 25 °C is shown in Figure 2a. The on-chip insertion loss is ~1.46 dB. Two group resonances with different FSRs are observed. The FSRs for different MRRs are FSR1 = 2.9 nm and FSR2 = 2.41 nm, respectively. We observe the resonance dips at 1546.00 nm and 1578 nm. The FSR for our dual-MRR is about 14.56 nm. Figure 2b shows the transmission of the resonator peak at 1560.56 nm. The full width at half maximum (FWHM) of the resonance peak for the present resonator is about Δλ = 463 pm at 1560.56 nm, indicating a loaded Q factor Q_load_ = λ/Δλ of 3370.

The temperature response of the sensor was investigated by placing it on a high-precision temperature console stage with a temperature resolution of 0.1 K. Figure 3a shows the measured spectra of the dual-MRR temperature sensor, indicating a linear wavelength shift with temperature from 0 K to 20 K. The resonant wavelength is extracted and linearly fitted as shown in Figure 3b. The slope of fitting results presents the temperature sensitivity, which is calculated as Δλ/ΔT = 85.74 pm/K with a correlation coefficient (R^2^) of 0.99996.

### 2.2. The Structure of AMZI

Mach–Zehnder Interferometers (MZIs), one of the most widely used structures, have been demonstrated in a wide range of applications, including wavelength division multiplexers [34,35], optical switches [36], electro-optical modulators [37] and biosensors [38,39]. In this paper, we also propose AMZI temperature sensors based on the p-Si platform. The microscopic image of AMZI is shown in Figure 4. It consists of two 3 dB couplers and two arms with different lengths. Directional coupler (DC), Y-branch splitter, and multi-mode interferences (MMI) are common structures for 3 dB couplers. Considering with the bandwidth and process tolerance, we choose MMI as 3 dB couplers for the AMZI sensors.

The 1 × 2 MMI used in the sensor is shown in Figure 5a. A series of cascaded MMIs were fabricated to characterize the insertion loss and uniformity of the output waveguides, as shown in Figure 5b. Transmission spectra from ports 1–7 in the range of 1460 nm to 1580 nm are shown in Figure 5c. Then, we fit the transmission loss at 1550 nm and found the slope of the linear fitting was −4.45, as shown in Figure 5d. It means the excess loss of MMI is 1.54 dB. The MMI shows superior uniformity according to the transmission spectra of ports 6 and 7.

The FSR is also important for an AMZI, which is given by
(1)$$\mathrm{FSR}=\frac{\lambda^{2}}{n_{g}\Delta L}$$
where λ is the center wavelength, n_g_ is the group index, and ∆L is the path length difference between the arms. The group index n_g_ is 4.11, which is calculated through finite difference eigenmode (FDE) method. The center wavelength λ is 1550 nm. The interferometer path length difference ∆L is 280 μm. Thus, the FSR of AMZI is 2.08 nm.

The same characterization system with a dual-MRR sensor is applied for the AMZI. The transmission spectrum of our designed AMZI sensor is shown in Figure 6 when the temperature is 25 °C. The AMZI shows an insertion of 16.5 dB. The FSR of AMZI is 2.06 nm, which is consistent with the calculated result.

We chose the resonant wavelength dip at 1549.07 nm to demonstrate the sensing application. The drift of the characteristic wavelength observes the temperature change. Figure 7a plots the spectrum change of AMZI when the temperature change varies from 0 K to 20 K. The extinction ratios of our designed AMZI’s resonant peak are almost constant with increasing temperatures. Figure 7b shows the linear fitting of the resonant wavelength changing with temperature changing. The sensitivity of the temperature sensor, which equals the slope of the straight line, is 0.0866 with a correlation coefficient (R^2^) of 0.998. Therefore, the AMZI sensor has a sensitivity of 86.66 pm/K.

## 3. Discussion

The performance comparison between temperature sensors on different platforms is listed in Table 1. In [40], a silicon-based dual-polarization MRR with a polyvinyl-alcohol (PVA) upper cladding is demonstrated for measuring humidity and temperature simultaneously. For TE and transverse magnetic (TM) polarization modes, the sensitivities of the sensor are 69.0 pm/K and 30.6 pm/K, respectively. Multifunctional sensors with humidity and temperature monitors are attractive, but sensitivity is lower than in our work. Low depth for the TM mode resonator is also a challenge for practical applications. To increase the sensitivity, fano resonance is achieved by introducing an air hole with a diameter of 368 μm into the center of the coupling [41]. However, the sensitivity is only 75.3 pm/K with an extinction ratio (ER) of 9.57 dB. Both of the MRR- and AMZI-based sensors demonstrated in this paper show a large ER of ~35 dB. However, the gap between the bus waveguide and ring of the sensor in [41] is 83 nm, which is hard for a MPW process. Replacing the silicon or silica waveguides with polymer waveguides is another effective method to improve the sensitivity. In [42], a MZI sensor with two arms consisting of hybrid waveguides providing the opposite temperature-dependent phase changes is demonstrated. One arm of the MZI sensor is narrowed to 40 nm, leaking the light to SU-8 cladding with a negative thermal optical coefficient (TOC). The opposite temperature dependent phase change enhances the sensitivity to 172 pm/K. In [43], a chip-scale temperature sensor with a high sensitivity of 228.6 pm/K based on a rhodamine 6G (R6G)-doped SU-8 whispering gallery mode microring laser is developed. However, polymer materials need an effective method to improve their long-term stability [44,45]. Using the same whispering gallery mode microring structure, a low sensitivity of 19.37 pm/K is measured, owing to low TOC of SiN [46]. Another effective method is using Michelson interferometer (MI) to improve the influence of light propagation by heater. A compact size of 120 μm × 80 μm temperature sensors with 113.7 pm/k sensitivity is achieved [47]. This structure is also useful for thermo-optic switches [48]. In our work, we demonstrate two types p-Si temperature sensors based on MRR and AMZI structures. The fabrication is well compatible with the CMOS fabrication technique. Low fabrication temperatures enable the p-Si sensor to be used in multilayer integrated optical circuits. The sensitivity is to the same degree as the c-Si sensor. With new material introduced and useful structure applied, the sensitivity could be improved sharply. Moreover, p-Si is a normal material for electronic integrated circuits. A fully integrated biosensing electronic–photonic system-on-chip (EPSoC) could be achieved using this platform [7,49,50].

## 4. Conclusions

In conclusion, we demonstrated two types of p-Si temperature sensors experimentally. The two types of sensors were dual-MRR and AMZI structures. The dual-MRR sensor had a large FSR of 14.56 nm and the Q_load_ of dual-MRR is 3370. We optimized the 1 × 2 MMI of the AMZI sensor with an excess loss of 1.54 dB and uniform spectral uniformity. For both sensors, we used the shift of the resonant wavelengths to calculate the amount of temperature change. The results show the sensitivity of the dual-MRR sensor and AMZI sensor are 85.74 pm/K and 86.6 pm/K, respectively. It confirms that the temperature sensors based on the p-Si waveguides have large temperature sensitivity and a compact footprint. Therefore, they show great potential to realize temperature monitoring of multilayer integrated optical circuits and EPICs.

## Figures and Tables

**Figure 1 sensors-22-09357-f001:**
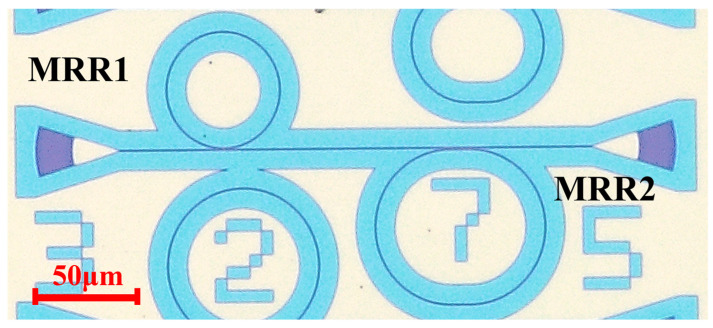
Microscopic image of the dual-MRR temperature sensor.

**Figure 2 sensors-22-09357-f002:**
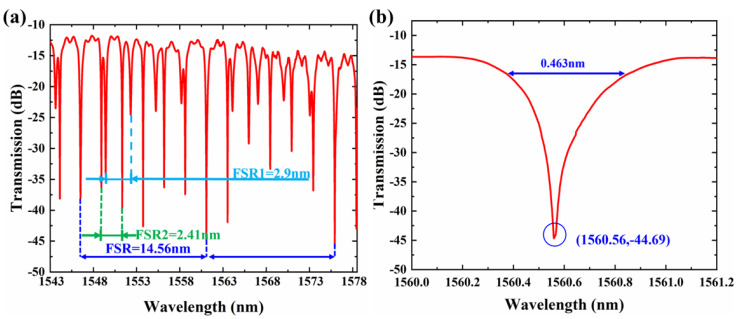
(**a**) The spectrum of the dual-MRR sensor. (**b**) The transmission of overlapping resonant peaks.

**Figure 3 sensors-22-09357-f003:**
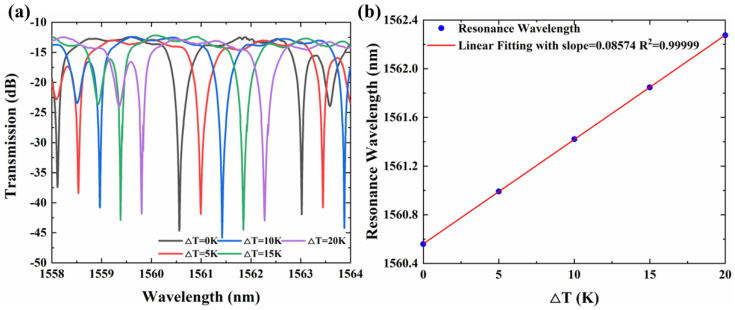
(**a**) Spectra of the dual-MRR sensor at different temperatures (ranging from 0 K to 20 K). (**b**) Overlapping resonant peak wavelength at different temperatures.

**Figure 4 sensors-22-09357-f004:**
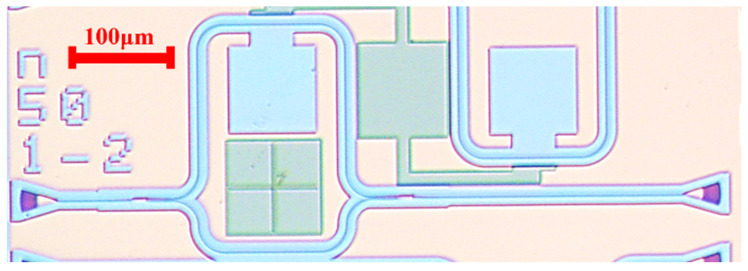
Microscopic image of the temperature sensor based on AMZI.

**Figure 5 sensors-22-09357-f005:**
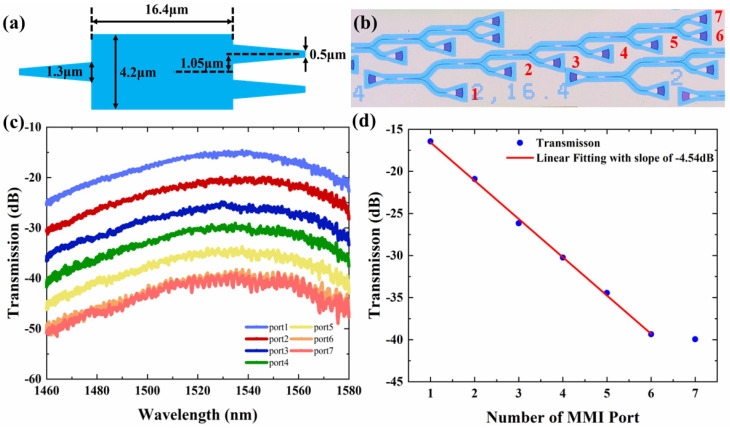
(**a**) Schematic of 1 × 2 MMI. (**b**) Microscopic image of the cascade MMI. (**c**) Measured trans-mission spectra of the cascaded 1 × 2 MMIs at the wavelength range of 1460–1580 nm. (**d**) Linear fitting of the normalized transmission at 1550 nm wavelength.

**Figure 6 sensors-22-09357-f006:**
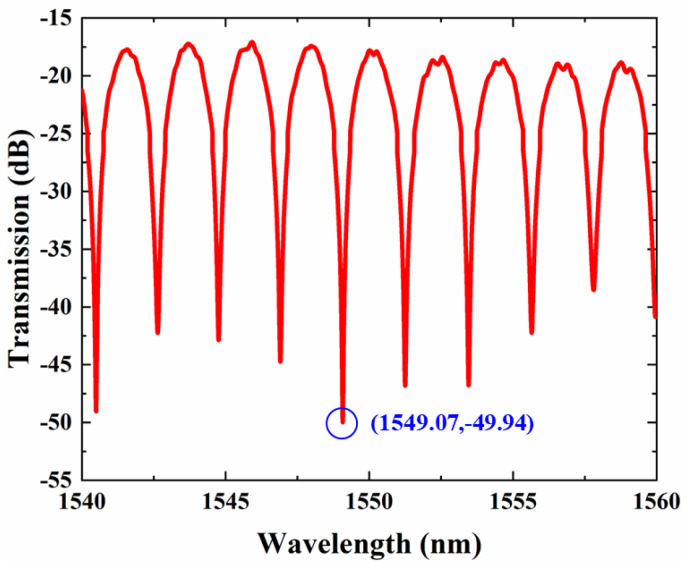
The spectrum of our designed AMZI.

**Figure 7 sensors-22-09357-f007:**
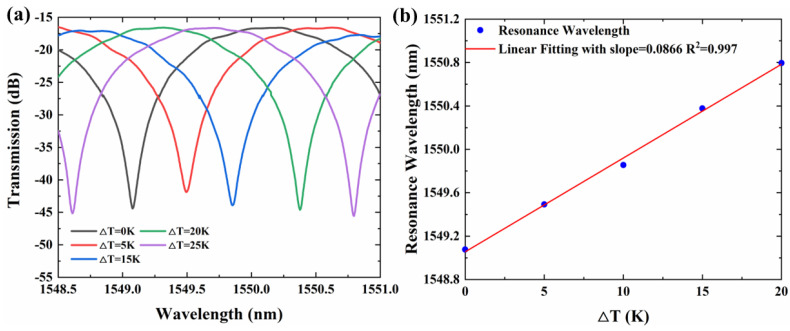
(**a**) Spectra of the AMZI sensor at different ambient temperatures. (**b**) Resonant peak wave-length at different temperatures and linear fitting.

**Table 1 sensors-22-09357-t001:** Comparison of temperature senors.

Reference	Waveguide Materails	Structure	Sensitivity(pm/K)	Radius/Footprint
[40]	c-Si	MRR	69.0 pm/K	20 μm
[41]	c-Si	MRR	75.3 pm/K	20 μm
[42]	c-Si	AMZI	172 pm/K	N.A.
[43]	R6G-SU-8	MRR	228.6 pm/K	110 μm
[46]	SiN	MRR	19.37 pm/K	100 μm
[47]	c-Si	MI	113.7 pm/K	120 μm × 80 μm
This work	p-Si	MRR	85.7 pm/K	30 μm
This work	p-Si	AMZI	86.6 pm/K	400 μm × 260 μm

## Data Availability

Not applicable.

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
