# Peer review of "Optical Temperature Sensor Based on Polysilicon Waveguides"

_sensors, 2022, doi:10.3390/s22239357_

Round 1
Reviewer 1 Report
This manuscript reported two kinds of temperature sensors based based on polysilicon (p-Si) material, using dual-microring resonator (MRR) and asymmetric Mach–Zehnder interferometer (AMZI), respectively. The results show a better sensitivity than most of the c-Si based sensors. Due to some critical issues not properly addressed, I would recommend its publication after revision. My comments and suggestions are as follows:
1. The schematic diagram of design structure, fabrication process and characterization system should be given in details for visualized description.
2. The waveguides of 500nm×220 nm was etched using one-step 70nm on depositing p-Si of 2 μm thickness. However, in fact, it is difficult to control the dimensions strictly. Could the authors give the measurement results of the actual dimension which are consistent with the performance experimental results?
3. A broadband tunable laser system was only mentioned in the manuscript, it would be beneficial if the authors can show the other components of the characterization system.
4. “The spectrum of the dual-MRR sensor from 1543 nm to 1568 nm at 25℃ is shown in Fig. 2 (a).” In this sentence, the spectrum range is inconformity with the Figure 2(a).
5. Why the Qload and the group index ng is 3370 and 4.11, respectively? The authors should give short explanations.
6. In Discussion, line No.157, the word ‘chanlenge’ is misspell.
Reviewer 2 Report
The authors describe a highly sensitive temperature sensor based on AMZIs and MMIs which are suitable for PICs. The results are impressive but a description of a possible application scenario is missing to complete that story. Next to that there are a few minor comments.
General: It would be very helpful for the reader if the authors can describe their characterization method / setup more detailed. How they achieve the coupling and transmission losses.
- Is there any other special equipment (spectrometer) next to the tunable laser required for the measurements? Can the authors mention possible applications where such an accurate temperature measurement is helpful.
Line 115-121: This part seems to be misplaced. There is a section about AMZIs, then without any introduction the part for the MMIs appears and continues with AMZIs. Better description and introduction of the MMIs is necessary.
I found no results of the characterization of the MMIs. Is the resolution good enough to distinguish the mentioned wavelength shift of 30 and 69pm/K, respectively.
Line 128: What’s the value of ΔL.
Line 130: 1550 nm.
Line 155: Have the values of 69pm/K and 30.6pm/K, respectively, been determined by the MMIs or with an additional spectrometer?
Line 156: sensor with humidity and temperature monitor is an attractive, but sensitivity is lower
than our work. – This sentence need to be completed.
Line 157: mode resonator is also challenging(?)
Line 167: thermos-optic
Line 176: This structure is also
Conclusion: A possible application of these technology is missing. Where can such a highly sensitive optical temperature sensor be used?
